# The Immunohistochemical Expression of Epithelial–Mesenchymal Transition Markers in Precancerous Lesions and Cervical Cancer

**DOI:** 10.3390/ijms24098063

**Published:** 2023-04-29

**Authors:** Aneta Popiel-Kopaczyk, Aleksandra Piotrowska, Patrycja Sputa-Grzegrzolka, Beata Smolarz, Hanna Romanowicz, Piotr Dziegiel, Marzenna Podhorska-Okolow, Christopher Kobierzycki

**Affiliations:** 1Division of Histology and Embryology, Department of Human Morphology and Embryology, Wroclaw Medical University, 50-368 Wroclaw, Poland; aleksandra.piotrowska@umw.edu.pl (A.P.); piotr.dziegiel@umw.edu.pl (P.D.); christopher.kobierzycki@umw.edu.pl (C.K.); 2Division of Anatomy, Department of Human Morphology and Embryology, Wroclaw Medical University, 50-368 Wroclaw, Poland; patrycja.sputa-grzegrzolka@umw.edu.pl; 3Department of Pathology, Polish Mother’s Memorial Hospital Research Institute, 93-338 Lodz, Poland; smolbea@wp.pl (B.S.); hanna-romanowicz@wp.pl (H.R.); 4Department of Physiotherapy, University School of Physical Education, 51-612 Wroclaw, Poland; 5Division of Ultrastructural Research, Wroclaw Medical University, 50-368 Wroclaw, Poland; marzenna.podhorska-okolow@umw.edu.pl

**Keywords:** SNAIL, TWIST, SLUG, EMT, cervical cancer, cervical intraepithelial neoplasia, immunohistochemistry, HSIL, LSIL

## Abstract

In the epithelial–mesenchymal transition (EMT) process, cells lose their epithelial phenotype and gain mesenchymal features. This phenomenon was observed in the metastatic phase of neoplastic diseases, e.g., cervical cancer. There are specific markers that are expressed in the EMT. The aim of this study was to determine the localization of and associations between the immunohistochemical (IHC) expression of TWIST, SNAIL, and SLUG proteins in precancerous lesions and cervical cancer. The IHC analysis disclosed higher expressions of EMT markers in precancerous lesions and cervical cancer than in the control group. Moreover, stronger expression of TWIST, SNAIL, and SLUG was observed in cervical intraepithelial neoplasia grade 3 (CIN3) vs. CIN1, CIN3 vs. CIN2, and CIN2 vs. CIN1 cases (*p* < 0.05). In cervical cancer, IHC reactions demonstrated differences in TWIST, SNAIL, and SLUG expression in grade 1 (G1) vs. grade 2 (G2) (*p* < 0.0011; *p* < 0.0017; *p* < 0.0001, respectively) and in G1 vs. grade 3 (G3) (*p* < 0.0029; *p* < 0.0005; *p* < 0.0001, respectively). The results of our study clearly showed that existing differences in the expression of the tested markers in precancerous vs. cancerous lesions may be utilized in the diagnosis of cervical cancer. Further studies on bigger populations, as well as in comparison with well-known markers, may improve our outcomes.

## 1. Introduction

Cervical cancer is one of the leading malignancies of the female genital system. In the United States alone, in 2023, over 13,960 women will be diagnosed with cervical cancer, and 4280 of them will die. There are still 341,831 deaths among women per year worldwide, despite reductions in the mortality and incidence of cervical cancer brought on by the human papilloma virus (HPV) vaccine and increased screening [1,2]. A squamous intraepithelial lesion is formed by the abnormal growth of squamous cells on the surface of the cervix. The degree of dysplasia is determined by the percentage of cervical epithelium that contains dysplastic cells. When compared to the more serious CIN2 and CIN3 (high-grade), which proceed to involve the full thickness of the epithelium, CIN1 (low-grade) only affects the lower one-third or less of the epithelium. When dysplasia penetrates the basement membrane, it develops into cancer. Cervical cancer is usually diagnosed between the ages of 35 and 44, with a reported average 5-year survival rate of 66% [3]. There are several factors that influence this value. One of them is the clinical stage at diagnosis. When diagnosed at an early stage, the 5-year survival rate is 92%; if the cancer has spread regionally to surrounding tissue and lymph nodes, then the rate is 58%, but it dramatically declines to 18% when distant metastases occur [4]. Patients with locally advanced cervical cancer (stage IB3 to IVA) have a higher rate of recurrence. The early stage of the disease limited to the cervix and uterus (stage IA to IB2) can be treated by radical surgery or concomitant chemotherapy, which is based on patient characteristics and the volume of the disease [5]. After surgery alone, the probability of relapse is at least 30% [6,7]. Although it is not common at initial diagnosis, metastases develop in 15% to 61% of women with cervical cancer, usually within the first two years after finishing treatment [8]. The presence of invasion and metastasis is the major cause of most cancer-related deaths. The epithelial–mesenchymal transition (EMT) process is closely related to tumor metastasis. During the process of EMT, polarized epithelial tumor cells gain invasive and migratory characteristics, leave the primary site, invade the basement membrane, intravasate into blood or lymph vessels, transport through the circulation, extravasate from the circulation, disseminate into a secondary site, and finally, grow at the metastatic site. EMT can be triggered by the dysregulation of oncogenes, tumor suppressors, miRNAs, and growth factor signals. Several transcription factors influence the process of EMT: i.a., twist family bHLH transcription factor 1 (TWIST), snail family zinc finger 1 (SNAIL), and snail family zinc finger 2 (SLUG) [9]. 

TWIST belongs to the helix–loop–helix transcription factors engaged in the EMT process. Its expression leads to a loss of E-cadherin-mediated cell–cell adhesion, activates mesenchymal markers, and initiates cell motility. Li et al. showed that the expression of TWIST is crucial for the activation of the β-catenin and Akt pathway in HeLa cells to maintain the EMT process [10]. High expression of TWIST was linked with chemo- and radiotherapy resistance [11,12,13]. Moreover, TWIST overexpression is associated with lower patient survival rates and cervical cancer progression [9,11,14,15,16]. 

The SNAIL family is composed of zinc-finger-containing transcription factors and includes SNAIL, SLUG, and SMUC. SNAIL is one of the most widely studied regulators of the EMT process. Its expression is controlled at many levels: transcriptional, translational, and post-translational [11]. As a transcriptional factor, SNAIL rules genes related to EMT-independent functions such as cell survival, motility, anti-apoptosis, immune suppression, stem cell properties, and chemo-resistance [17,18,19,20,21]. 

SLUG is a member of the SNAIL superfamily and has a pivotal role in the EMT process. Increased expression of SLUG can lead to reduced E-cadherin expression and the onset of EMT. There are reports in the literature that SLUG initiates EMT and promotes metastasis through its trans-repression effect on E-cadherin regulation in cervical cancer [22,23]. Xian Liu et al. showed that exogenously expressed SLUG in HeLa and SiHa cells significantly enhanced cell motility in vitro and promoted distant metastasis in vivo [24]. On the other hand, Nan Cui et al. demonstrated that SLUG acts as a suppressor gene, inhibiting the proliferation of cervical cancer in vitro and tumor formation in vivo [25]. The scientific findings presented above indicate a strong need to expand research on EMT marker expression in cervical cancer.

The aim of this study was, thus, to determine the localization of and associations between immunohistochemical (IHC) expression of the TWIST, SNAIL, and SLUG proteins in precancerous lesions and cervical cancer.

## 2. Results

We determined the IHC expression of TWIST, SNAIL, and SLUG in 124 cervical cancer cases, 229 CIN cases, and 145 patients in the control group. We observed interesting cellular expression patterns. For TWIST, cytoplasmic localization in cancer (Figure 1E), CIN lesions (Figure 1B–D), and normal tissue (Figure 1A) were found. The expression of the SNAIL protein in normal tissue (Figure 2A) was nuclear, but in cervical cancer (Figure 2E) and CIN lesions (Figure 2B–D), the reaction was nuclear–cytoplasmic. The expression of the SLUG protein was nuclear–cytoplasmic in cancer cells (Figure 3E) and nuclear in CIN lesions (Figure 3B–D) and normal tissue (Figure 3A).

### 2.1. Precancerous Lesions

The highest TWIST, SNAIL, and SLUG expression was noted in CIN3 and was significantly higher as compared to that in CIN1 and CIN2 (respectively, TWIST *p* < 0.0001, SNAIL *p* < 0.0013; SLUG *p* < 0.0001, Figure 4A; Mann–Whitney test). Moreover, expression was significantly higher in the high-grade squamous intraepithelial lesion (HSIL) group than in the low-grade squamous intraepithelial lesion (LSIL) group (*p* < 0.0001; Mann–Whitney test; Figure 4B). In addition, significantly stronger expression of TWIST, SNAIL, and SLUG was observed in CIN3 vs. CIN1, CIN3 vs. CIN2, and CIN2 vs. CIN1 cases (*p* < 0.05 for all; Dunn’s multiple comparison tests). Moreover, the Spearman correlation test revealed a significant correlation between EMT markers in CIN lesions (Table 1; Figure 5A–C).

### 2.2. Cervical Cancer

In cervical cancer cases, the expression of EMT markers was higher than in the control group (*p* < 0.0001 for TWIST, SNAIL, and SLUG; Mann–Whitney test; Figure 6A). Moreover, we demonstrated a considerable difference in TWIST, SNAIL, and SLUG expression between G1 and G2 (respectively, *p* < 0.0011, Figure 6B; *p* < 0.0017, *p* < 0.0001, Mann–Whitney test), as well as between G1 and G3 (accordingly, *p* < 0.0029, Figure 6B; *p* < 0.0005, *p* < 0.0001; Mann–Whitney test). Between G2 and G3, we found no significant differences in the expression of EMT markers (TWIST: *p* < 0.8304, Figure 6B; SNAIL: *p* < 0.1208, SLUG: *p* < 0.7947; Mann–Whitney test). The Spearman correlation test showed that the expression of TWIST positively correlated with the expression of the SNAIL protein in cervical cancer (r = 0.4338, *p* < 0.0001; Figure 7C; Table 1). A positive correlation was also shown between SLUG protein expression and SNAIL (r = 0.3764, *p* < 0.0001; Figure 7A; Table 1). There were no significant correlations between SLUG protein expression and TWIST protein expression in cervical cancer (r = 0.1066, *p* < 0.2483; Figure 7B; Table 1). The expression of all EMT markers in the LSIL, HSIL, G1, G2 and G3 was significantly different in Kruskal–Wallis test (Figure 8; *p* < 0.0001). Dunn’s multiple comparison test showed statistically significant difference of TWIST expression between groups: LSIL vs. HSIL, LSIL vs. G1, LSIL vs. G2, LSIL vs. G3; SNAIL expression were different in groups: LSIL vs. HSIL, LSIL vs. G1, LSIL vs. G2, LSIL vs. G3, HSIL vs. G2, HSIL vs. G3 and the SLUG expression were statistically different in the groups: LSIL vs. HSIL, LSIL vs. G1, LSIL vs. G2, LSIL vs. G3, HSIL vs. G1, HSIL vs. G2, HSIL vs. G3, G1 vs. G2, G1 vs. G3. 

## 3. Discussion

The epithelial–mesenchymal transition phenomenon plays an important role in cervical cancer progression. Many studies were conducted to investigate the mechanism of the progression of cervical cancer from CIN. The formation of metastases from cervical cancer is a process that involves multiple steps and a cascade of reactions. The prognosis significantly decreases when distant metastases occur because the treatment of local lesions is more effective than systemic therapy [26]. The cadherins are key components that contribute to cell motility and invasiveness via EMT. A reduction in E-cadherin expression leads to a loss of cell polarity and decreased cell adhesion [27]. In many human malignancies, downregulation of E-cadherin is associated with a poor prognosis and is a key feature of cancerogenesis, such as tumor spreading [28,29]. Several important genes induce EMT and act as E-cadherin repressors, such as SNAIL, SLUG, and TWIST. The risk of developing invasive cervical cancer from HSIL is approximately 20% (10–40% according to the literature) [30,31,32]. Previous studies described the expression of EMT markers in cervical cancer; however, there were no prior data on their expression in preinvasive lesions. Due to the lack of such studies, we decided to examine whether there were any differences in EMT marker expression between precancerous lesions, cervical cancer, and normal cervical epithelium. We showed that the expression of SLUG, SNAIL, and TWIST was significantly higher in CIN lesions than in the control group. In addition, we demonstrated that EMT marker expression changed with the histological stage and rose with the stage (TWIST: *p* < 0.0001, Figure 4; SNAIL: *p* < 0.0013, Figure 4; SLUG: *p* < 0.0001, Figure 4; Mann–Whitney test). This may suggest that EMT has a role in the progression of CIN lesions into cervical cancer. Several studies suggested the involvement of SNAIL, SLUG, and TWIST in the development of cervical cancer [15,33,34,35]. The analysis of IHC expression showed higher expression of SLUG, SNAIL, and TWIST in cervical cancer than in the control group. Tian et al. showed that high SNAIL expression predicts a lower survival rate and is correlated with highly aggressive FIGO stage and LNM (lymph node metastasis) status in cervical cancer patients [36]. The overexpression of SLUG observed in cervical cancer is consistent with the work of Liu et al., who revealed a significant effect of SLUG on the EMT process [24]. In clinical practice, markers of EMT play an increasingly important role and are crucial for many treatments of cervical cancer. Dai et al. described novel therapeutic strategies based on negative regulation of the Wnt signaling pathway and reversing the EMT process. HMQ-T-F2 (F2) was shown to suppress the expression of SNAIL [37]. Due to EMT-induced anti-apoptotic abilities, enhanced DNA damage repair, and a changed drug metabolism route, tumor cells become resistant to therapy and cytotoxicity [38]. Tumor radiation sensitivity rapidly declines as EMT advances because EMT is inextricably linked to tumor radiation resistance. Increased radiation resistance in cervical cancer may be caused by TRIP4 overexpression in tumor tissues and cancerous cells, which may encourage EMT and activate the PI3K/Akt and MAPK/ERK signaling pathways [39]. XAV939 is an inhibitor of the Wnt signaling pathway that was shown to increase cervical cancer cells’ radiation sensitivity [40,41]. The expression of EMT markers seems to be an important aspect in planning the treatment of patients and predicting the response to treatment. Work on finding efficient therapeutic targets for cervical cancer metastasis seems to be justified.

## 4. Materials and Methods

This study was performed on selected archival paraffin-embedded specimens. Patients were operated on between 2014 and 2017 at the Polish Mother’s Memorial Hospital in Lodz. The patients, aged from 25 to 86 years old, were of the female sex. The control group consisted of normal cervical tissue obtained from patients who underwent total hysterectomy due to uterine leiomyomas. The study group consisted of CIN1 (31), CIN2 (75), CIN3 (123), and cervical cancer (124) cases, whereas the control group was composed of 145 cases. The study was approved by the Ethics Committee of Medical University in Wroclaw (21 December 2022; protocol code 1003/2022).

### 4.1. TMA Construction

To confirm the histological diagnosis and determine whether the material was suitable for further examination, 6 m thick paraffin sections were produced and stained with hematoxylin and eosin (HE). Briefly, slides were scanned using the histologic Pannoramic MIDI scanner (3DHistech Ltd., Sysmex Suisse AG, Horgen, Switzerland). The sites of CIN in the altered cervix epithelium were then selected and digitally tagged after two independent pathologists reviewed the scans. Next, duplicate tissue core punches (2 mm) for each case were taken from the appropriate paraffin donor blocks for use in the preparation of TMAs (TMA Grand Master; 3DHistech). Normal epithelial tissue from the cervix was designated as the control group.

### 4.2. Immunohistochemistry (IHC)

All IHC reactions were performed on 4 µm thick paraffin slides from TMA using a Dako Autostainer Link48 (Dako, Glostrup, Denmark). The following primary antibodies were used: SLUG (1:50, sc-166476, Santa Cruz Biotechnology, Santa Cruz, CA, USA), TWIST (1:50, ab-50887, Abcam, Cambridge, UK), and SNAIL (1:400, 13099-1-AP, Proteintech, Rosemont, IL, USA). All procedures were conducted as previously described [42]. EnVision FLEX (Dako) was used for the visualization of antibodies, in accordance with the manufacturer’s instructions. Cytoplasmic reactions of SLUG, SNAIL, and TWIST were evaluated via Pannoramic Viewer Digital (3DHistech) image analysis and the routinely used immunoreactive scale (IRS) by Remmele and Stegner. This scale evaluates the percentage of positive cancer cells (A) and the staining intensity of the reaction (B). The final result is the product of these two values (AxB). The nuclear expression of SNAIL and SLUG was evaluated semi-quantitatively based on the percentage of positively stained cells of the whole section (3 slides per case) and encoded as follows: 0: absence of staining; 1: 1–10% cells stained; 2: 11–25% cells stained; 3: 26–50% cells stained; and 4: over 50% cells stained. 

### 4.3. Statistical Analysis

All statistical analyses were performed using GraphPad Prism 5.0 software (GraphPad, La Jolla, CA, USA) with Spearman correlation, Kruskal–Wallis, Dunn’s multiple comparison, and Mann–Whitney tests. *p*-values < 0.05 were considered to be statistically significant. The image GP program was used to create the diagrams [43].

## 5. Conclusions

In our study, the expression of TWIST, SNAIL, and SLUG increased gradually as lesions progressed from LSIL to HSIL. We are the first to show a gradual increase in EMT markers in CIN lesions. Moreover, we confirmed a higher expression of TWIST, SNAIL, and SLUG in cervical cancer than in the control group. The aforementioned data support the idea that EMT elements play a role in the development of cancer.

## Figures and Tables

**Figure 1 ijms-24-08063-f001:**
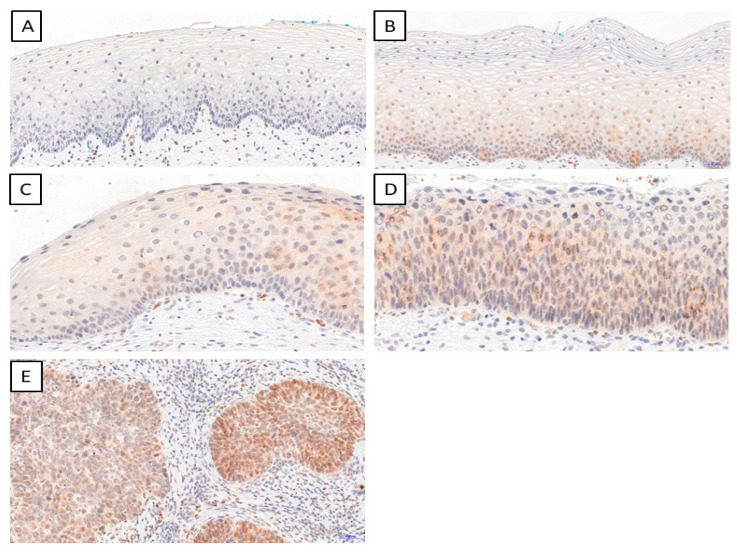
The IHC expression of TWIST (normal cervical tissue (**A**); CIN1 (**B**); CIN2 (**C**); CIN3 (**D**); cervical cancer (**E**)); magnification ×200.

**Figure 2 ijms-24-08063-f002:**
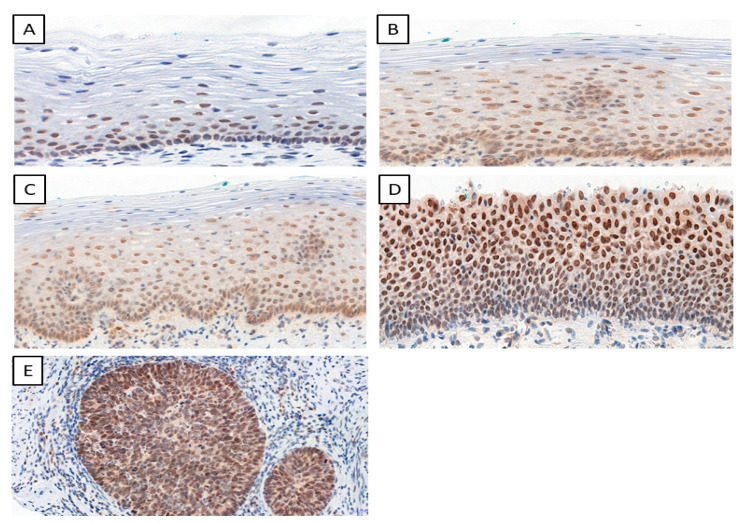
The IHC expression of SNAIL (normal cervical tissue (**A**); CIN1 (**B**); CIN2 (**C**); CIN3 (**D**); cervical cancer (**E**)); magnification ×200.

**Figure 3 ijms-24-08063-f003:**
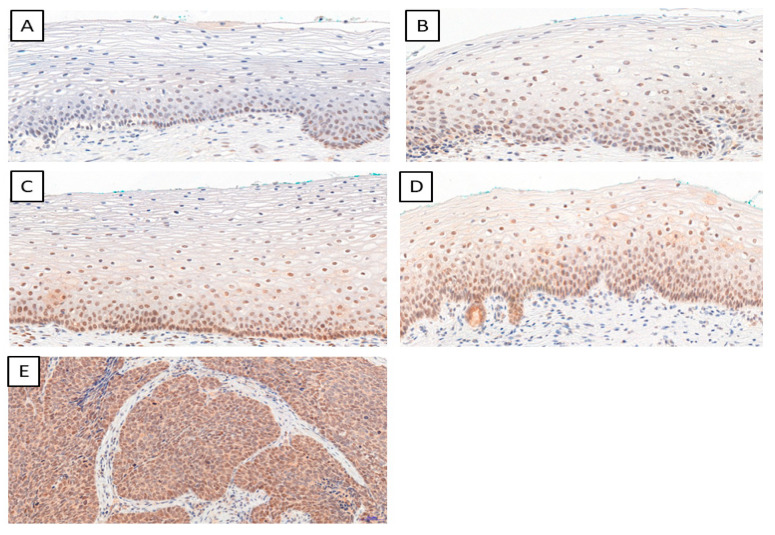
The IHC expression of SLUG (normal cervical tissue (**A**); CIN1 (**B**); CIN2 (**C**); CIN3 (**D**); cervical cancer (**E**)); magnification ×200.

**Figure 4 ijms-24-08063-f004:**
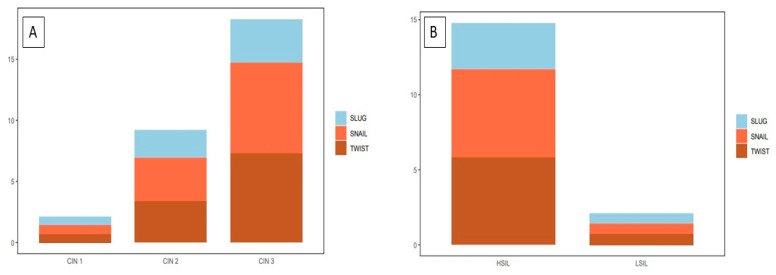
A comparison of the IHC expression of the tested markers in precancerous lesions: (**A**) TWIST expression in CIN1, CIN2, and CIN3; SNAIL expression in CIN1, CIN2, and CIN3; SLUG expression in CIN1, CIN2, and CIN3); (**B**) TWIST expression in HSIL and LSIL; SNAIL expression in HSIL and LSIL; SLUG expression in HSIL and LSIL Mann–Whitney test).

**Figure 5 ijms-24-08063-f005:**
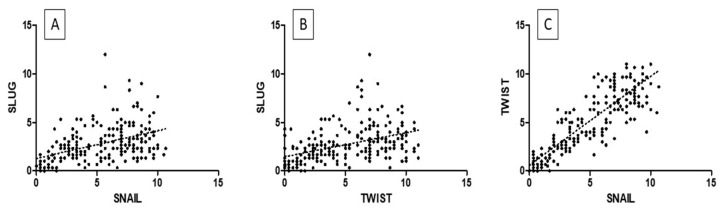
The correlations between IHC expression levels in precancerous lesions: (**A**) SLUG vs. SNAIL; (**B**) SLUG vs. TWIST; (**C**) TWIST vs. SNAIL (Spearman correlation test).

**Figure 6 ijms-24-08063-f006:**
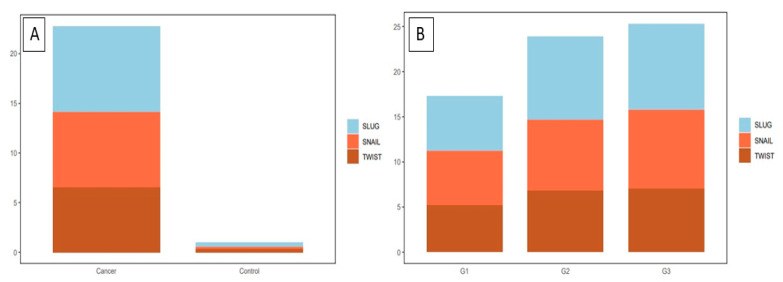
The IHC expression of the tested markers in cervical cancer: (**A**) TWIST expression in cervical cancer and control group; SNAIL expression in cervical cancer and control group; SLUG expression in cervical cancer and control group (Mann–Whitney test). (**B**) A comparison of the IHC expression of TWIST, SNAIL, and SLUG in regard to the histological malignancy G (Mann–Whitney test).

**Figure 7 ijms-24-08063-f007:**
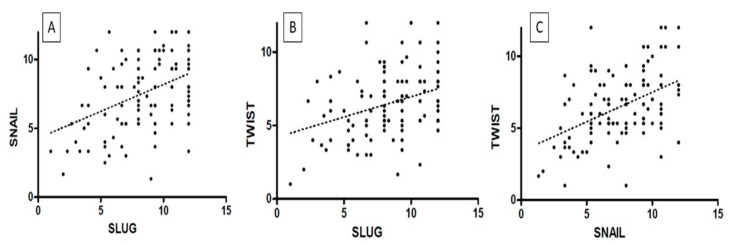
The correlations between IHC expression levels in cervical cancer: (**A**) SLUG vs. SNAIL; (**B**) SLUG vs. TWIST; (**C**) SNAIL vs. TWIST (Spearman correlation test).

**Figure 8 ijms-24-08063-f008:**
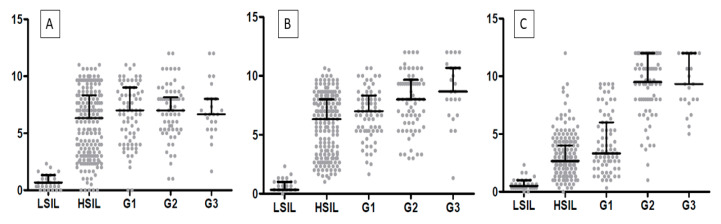
A comparison of the IHC expression of (**A**) TWIST, (**B**) SNAIL, and (**C**) SLUG in regard to the LSIL, HSIL, and histological malignancy G (*p* < 0.05, Mann–Whitney test).

**Table 1 ijms-24-08063-t001:** Spearman correlation test results. All correlations except SLUG vs. TWIST in cervical cancer were statistically significant.

	CIN		Cervical Cancer	
	SLUG	TWIST	SLUG	TWIST
SNAIL	r = 0.4787 *p* < 0.0001	r = 0.8470 *p* < 0.0001	r = 0.3764 *p* < 0.0001	r = 0.4338 *p* < 0.0001
TWIST	r = 0.2157 *p* < 0.0021	NA	r = 0.1066 *p* < 0.2483	NA

## Data Availability

The data presented in this study are available on request from the corresponding author. The data are not publicly available due to privacy issues.

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
