# Peer review of "The Immunohistochemical Expression of Epithelial–Mesenchymal Transition Markers in Precancerous Lesions and Cervical Cancer"

_ijms, 2023, doi:10.3390/ijms24098063_

Round 1
Reviewer 1 Report
The title and topic of the article is very interesting and very important.
There are several important genes that induce EMT and act as E-cadherin repressors such as SNAIL, SLUG and TWIST. As the authors indicate in the article, the risk of developing invasive cervical cancer due to HSIL is about 20%.
The authors, analyzing studies in the world, noticed that there are already studies describing the expression of EMT markers in cervical cancer, but there is no data on their expression in pre-invasive lesions. Therefore, they decided to check whether there are differences in the expression of EMT markers between precancerous lesions, cervical cancer and normal cervical epithelium. In this paper, they showed that the expression of SLUG, SNAIL and TWIST is significantly higher in CIN lesions than in the control group. In addition, they showed that the expression of the EMT marker varies with histological stage and increases with stage.
However, I have a few minor questions and comments (which do not necessarily affect the quality of the work presented).
1. Statistics show that cervical cancer is usually diagnosed between the ages of 35 and 44. Did the authors pay attention to this in their research? Because the samples they had were taken from patients between the ages of 25 and 86.
2. Immunohistochemical expression of TWIST, SNAIL and SLUG is shown in Figure 6. Nowhere in the text is there a description for this drawing.
3. Are all data in table 1?
Is it not necessary to provide an individual CIN correlations?
4. On page 1 line 14 there is no valid e-mail address.
5. On page 2, line 93, there are two dots at the end of the sentence.
6. Descriptions need to be unified. Sometimes it's Table 1, sometimes it's table 1. Sometimes it's Figure, sometimes it's figure (page 3 and 4).
7. I would increase the size of Figures (1, 2, 3, 4 and 5).
Author Response
Dear Reviewer,
Thank you very much for all your valuable suggestions and comments. Please find the point-by-point response below.
- Statistics show that cervical cancer is usually diagnosed between the ages of 35 and 44. Did the authors pay attention to this in their research? Because the samples they had were taken from patients between the ages of 25 and 86.
Thank you for your comment. The study group included: 7 patients in age group <35 yo; 51 patients in age group 35-45 and 87 patients had more than 45 yo. A slight dominance of the oldest group may indicate late diagnosis of cervical cancer poor patient education about prophylaxis.
- Immunohistochemical expression of TWIST, SNAIL and SLUG is shown in Figure 6. Nowhere in the text is there a description for this drawing. Thank you for your comment. We have added a description.
- Are all data in table 1?
Is it not necessary to provide an individual CIN correlations? Thank you for your suggestions. All data from correlation tests are in the table.
- On page 1 line 14 there is no valid e-mail address. Thank you. We have corrected it.
- On page 2, line 93, there are two dots at the end of the sentence. Thank you. We have corrected it.
- Descriptions need to be unified. Sometimes it's Table 1, sometimes it's table 1. Sometimes it's Figure, sometimes it's figure (page 3 and 4). Thank you. We have corrected it.
- I would increase the size of Figures (1, 2, 3, 4 and 5). Thank you. We have corrected it.
Reviewer 2 Report
The manuscript presented by the authors provides a comprehensive study on the localization and association between immunohistochemical expression in precancerous lesions and cervical cancer, using a large and valuable dataset. The study is important in deepening our understanding of cervical cancer. However, there are some issues for improvement in the manuscript.
Major reviews:
l It is wired that the authors provided Figure 6, which shows the IHC expression in cervical cancer and CIN3, but did not provide a description or explanation in the text. The authors should provide more information about this figure in the text. Additionally, it would be helpful if the authors could provide more pictures of the morphology variations of CIN1, CIN2, CIN3, cervical cancer, and control, which would provide more information for readers interested in the development of cervical cancer.
l It would be useful if the authors could add comparison data of CIN1, CIN2, CIN3, HSIL, LSIL, and different stages of cervical cancer as additional figures. This would help readers to better understand the differences and similarities between these conditions.
l The data visualization in poor. Please make them better. Such as barplot with error is not a good choice. Please change to boxplot with jitter (each sample in dots). Many webserver can easy plot beautiful and meaningful figures, such as ImageGP https://doi.org/10.1002/imt2.5.
Minor reviews:
l The authors should provide the full name of abbreviations, such as CIN, HSIL, LSIL, G1, G2 and G3, in the first time they are presented in the manuscript. This will help readers to understand the meaning of these abbreviations.
l The authors need to unify their expression of IHC expression in the figure notes and text. In some figures, they used abbreviations, while in others, they used full names. The authors should also use the same expression consistently in their text.
l In the Introduction, the authors provided statistics on the number of women who will be diagnosed with cervical cancer in 2023 in the US. However, they did not cite their data source. It is important to provide a reference for this statistic.
l There are some errors in the manuscript, such as an extra dot in line 93 and a grammar error in line 97. The authors should carefully proofread their manuscript to avoid such errors.
l The authors should unify the expression of p-values in the manuscript. They should choose one style, either P-values or lowercase p, and use it consistently throughout the manuscript. If they choose the lowercase p, it should also be italicized.
Author Response
Dear Reviewer,
Thank you very much for all your valuable suggestions and comments. The article was edited by the MDPI English Editing Team.
Please find point-by-point responses below.
The manuscript presented by the authors provides a comprehensive study on the localization and association between immunohistochemical expression in precancerous lesions and cervical cancer, using a large and valuable dataset. The study is important in deepening our understanding of cervical cancer. However, there are some issues for improvement in the manuscript.
Major reviews:
l It is wired that the authors provided Figure 6, which shows the IHC expression in cervical cancer and CIN3, but did not provide a description or explanation in the text. The authors should provide more information about this figure in the text. Additionally, it would be helpful if the authors could provide more pictures of the morphology variations of CIN1, CIN2, CIN3, cervical cancer, and control, which would provide more information for readers interested in the development of cervical cancer.
Thank you for your valuable comment. We have added more pictures of the morphology variations of CIN1, CIN2 and CIN3. We have added references to all figures in the text.
l It would be useful if the authors could add comparison data of CIN1, CIN2, CIN3, HSIL, LSIL, and different stages of cervical cancer as additional figures. This would help readers to better understand the differences and similarities between these conditions.
Thank you for this comment. We have added additional figures as you suggest.
l The data visualization in poor. Please make them better. Such as barplot with error is not a good choice. Please change to boxplot with jitter (each sample in dots). Many webserver can easy plot beautiful and meaningful figures, such as ImageGP https://doi.org/10.1002/imt2.5.
Thank you for your suggestions. We have added the new plot as boxplot with jitter.
Minor reviews:
l The authors should provide the full name of abbreviations, such as CIN, HSIL, LSIL, G1, G2 and G3, in the first time they are presented in the manuscript. This will help readers to understand the meaning of these abbreviations.
Thank you for your comment. We have provided the full names of these abbreviations.
l The authors need to unify their expression of IHC expression in the figure notes and text. In some figures, they used abbreviations, while in others, they used full names. The authors should also use the same expression consistently in their text.
Thank you for your comment. We have unified it.
l In the Introduction, the authors provided statistics on the number of women who will be diagnosed with cervical cancer in 2023 in the US. However, they did not cite their data source. It is important to provide a reference for this statistic. Thank you for your comment.
We have added the reference to these statistics.
Round 2
Reviewer 2 Report
The manuscript presented by the authors provides a comprehensive study on the localization and association between immunohistochemical expression in precancerous lesions and cervical cancer, using a large and valuable dataset. The study is important in deepening our understanding of cervical cancer. However, there are some issues for improvement in the manuscript.
Major reviews:
The manuscript has nine figures. Too many results in order to no important finding to be remembered by readers. At present, it like an experimental report, not a logical paper. The following suggestions will improve the structure of this paper:
1. Compact the important result into 3-4 figures according to the subtitle of results. Similar topics combo into one figure. Some not important figure can as a supplementary figure.
2. The figure is too plain, hard to attract the reader interest. Please replot all the figure by ImageGP (https://doi.org/10.1002/imt2.5) in colorful. Adjust the detail to scientific, clearly and readable, and attractive.
Author Response
Dear Reviewer,
Many thanks for your time and prompt review. We have corrected the figures according to your suggestions to make them colorful and interesting for the reader. We have tried to include the most important results, but we suggest leaving the figures with IHC staining separate so that they are readable by the viewer.
Round 3
Reviewer 2 Report
The author's response has answered my suggestion. I agree with the publication of this article.